# Fungal and Bacterial Communities in *Tuber melanosporum* Plantations from Northern Spain

Celia Herrero de Aza [1,2,*], Sergio Armenteros [2], James McDermott [3], Stefano Mauceri [4], Jaime Olaizola [5], María Hernández-Rodríguez [1,5] and Olaya Mediavilla [1,5]

1. Sustainable Forest Management Research Institute, University of Valladolid & INIA.Escuela Técnica Superior de Ingenierías Agrarias, University of Valladolid, Avda de Madrid 44, 34071 Palencia, Spain; maria@idforest.es (M.H.-R.); olaya@idforest.es (O.M.)
2. Research, Development and Innovation Department, ECM Ingeniería Ambiental, C. Curtidores 17, 34003 Palencia, Spain; sergio.armenteros.quirce@outlook.com
3. College of Science and Engineering, National University of Ireland, University Road, H91 TK33 Galway, Ireland; james.mcdermott@nuigalway.ie
4. UCD Michael Smurfit Graduate Business School, Carysfort Avenue, Blackrock, A94 D04 Dublin, Ireland; stefano.mauceri@ucdconnect.ie
5. Research, Development and Innovation Department, IDForest-Biotecnología Forestal Aplicada, Calle Curtidores 17, 34004 Palencia, Spain; jaime@idforest.es
* Correspondence: celia.herrero.aza@uva.es; Tel.: +34-979-108-316

**Abstract:** *Tuber melanosporum* (Ascomycota, Pezizales) is an ectomycorrhizal fungus that produces highly appreciated hypogeous fruiting bodies called black truffles. The aim of this paper was to research the composition of ectomycorrhiza-associated fungal and bacterial communities in *T. melanosporum* oak plantations. Results of this paper showed the competitive effect of *T. melanosporum* on other fungal species, especially other mycorrhizal and pathogenic species. *T. melanosporum* was shown to be associated mainly with bacteria, some of them important for their properties as mycorrhizal helper bacteria. A dendrogram analysis of co-occurrence showed that *T. melanosporum* tended to co-occur with the following bacteria species: *Singulisphaera limicola*, *Nannocistis excedens* and *Sporosarcina globispora*. In addition, it was linked to fungal species such as *Mortierella elongata*, *M. minutissima*, *Cryptococcus uzbekistanensis*, *C. chernovii* and *C. aerius*. This study provides an exhaustive analysis of the diversity, structure and composition of fungal and bacterial communities associated with *T. melanosporum* to enhance understanding of the biology, composition and role of these communities in truffle plantations.

**Keywords:** black truffle; *Quercus ilex*; microbial diversity; plant-associated microorganisms; fungal and bacterial networks; metagenomics techniques

## 1. Introduction

*Tuber melanosporum* Vittad. is an ectomycorrhizal ascomycete producing hypogeous edible fruiting bodies. These fruiting bodies, commonly known as 'black truffles', attain a high commercial value around the world for their distinctive aroma [1], reaching soaring prices in the markets [2].

*Tuber melanosporum* naturally occurs in symbiosis with several oak species and hazelnut trees [1]. As a highly appreciated forest product, its cultivation has been attempted since early times. Truffle cultivation was achieved for the first time in France during the 19th century [3]. Ever since significant progress has been made in truffle cultivation techniques [4], however truffle cultivation is still not fully understood [5]. Wild truffle production has declined globally, while truffle orchards have been established worldwide [6], from the Mediterranean to all Mediterranean-climate regions around the world [7]. As a consequence, nowadays, most of the global black truffle production occurs in planted orchards, holm oak (*Quercus ilex* L.) being the most extensively used host species [8].

Spain is one of the most important black truffle producers in the world. Here, plantations with truffle-inoculated seedlings have increased in recent decades due to the fact that the production of truffles provides higher economic returns than any other forest product in many Mediterranean woods [9].

Truffle production is largely unpredictable and varies greatly among plantations, and from year to year. Thus, a more complete understanding of the truffle life cycle and of the main factors that influence fruiting body formation is needed in order to decrease uncertainty [10]. On a broad scale, several environmental factors are known to influence black truffle productivity, such as soil characteristics and climate [11,12]. However, in order to promote truffle cultivation, it is essential to have a deeper knowledge of the biotic factors that may favor the ectomycorrhizal establishment and truffle maturation [13]. Understanding the ecology of *T. melanosporum* and the inter- and intraspecific interactions in the soil is needed to improve production [5]. Ectomycorrhizal fungi form a symbiosis with trees and certain shrub species. They influence nutrient and water uptake and absorption [14], plant growth and survival [15,16], and plant resistance to plant pathogens [17]. Bacterial communities are the third partner in the symbiosis between fungi and plant roots [18,19], which seem to be crucial in the complex biological processes of exchange involving nutrients [20,21]. Bacteria are ubiquitous microorganisms that can play different symbiotic roles in plants and fungi [22]. Some species contribute to nutrient cycling, such as nitrogen, carbon and phosphorus [23]. In addition, it has been shown that bacteria can promote the establishment of ectomycorrhizal symbiosis [20,24], and some bacterial species are involved in the production of truffles [25]. Soil microorganisms form complex inter-species networks which play a key role in the structure and functions of the ecosystem [26,27]. It has been observed that the presence, abundance and dynamics of T. melanosporum in soil depend on the composition of the whole plant community [28]. Thus, truffle orchards provide a particularly interesting case for studying soil microorganisms, as the high metabolic activity of black truffle mycelium likely influences other fungal and bacterial soil species.

In order to have a better understanding of soil ecosystem processes, it is essential to address the fungal and bacterial community at the same time [29]. In fact, the diversity of the microorganisms is commonly used as a bioindicator of ecosystem quality [30]. We hypothesized that assessing fungal and bacterial communities linked to truffle plantations would allow us to acquire knowledge on the relationships between different species and community organizations and interactions and to find indicator species associated with truffle production.

Our main objective was to provide a better knowledge of the soil microbiome (comprising fungi and bacteria) found in truffle-inoculated holm oak orchards, aiming to detect species that could be associated with black truffle mycelium. Thus, the specific objectives were: (1) to assess the fungal and bacterial richness and composition of a truffle plantation, and (2) to analyse the relationships between *T. melanosporum* and the rest of the microorganisms in the soil.

## 2. Materials and Methods

### 2.1. Data Sampling

Ten soil samples were collected at different truffle plantations in Northern Spain in October. The chosen truffle plantations were representative of the northeast of the Castilla y León region (Palencia, Burgos and Soria provinces). The age of the orchards at the time of sampling varied from 5 to 12 years. Despite the young age, truffles had been harvested at all orchards. Altitudes varied between 700 and 1100 m a.s.l. The pH values ranged between 8.1 and 8.3. The areas were characterized by a Mediterranean-Continental climate, with a xeric moisture regime, with a period of summer drought and rainfall, between 400 or 650 mm, with a maximum during autumn and spring. Exact locations of truffle orchards are not provided in order to ensure owners' privacy.

Soil samples were extracted using a cylindrical (2 cm radius, 20 cm deep, 250 cm$^3$) soil borer. A random sample of productive trees was chosen. A composite sample of

4 cores was made, taken in each of the 4 cardinal points of the selected trees, close to the trees (within approximately 1 m) and within the brûlé (the area devoid of vegetation around the symbiotic plants, where the fruiting bodies of *T. melanosporum* are usually collected) whenever it was visible. Samples were dried at room temperature and coarse elements were discarded. Once dry, samples were sieved (1 mm mesh) and homogenised. DNA was extracted from 0.25 g of dry soil per sample using Qiagen, Hilden, Germany, DNeasy PowerSoil Kit (Cat No./ID: 12888-100) at IDForest facilities and according to the manufacturer's instructions.

All the libraries were prepared following the two-step PCR Illumina protocol and these were subsequently sequenced on Illumina MiSeq instrument (Illumina, San Diego, CA, USA) using $2 \times 301$ paired-end reads. The two-step polymerase chain reactions (PCR) were used, which are convenient methods to generate amplicon libraries for Illumina sequencing. In the first PCR reaction, the targeted DNA region was amplified using specific primers flanked by tails. These tails allowed a second PCR reaction to add Illumina adaptor sequences and indexes to multiplex samples (https://www.ncbi.nlm.nih.gov/pmc/articles/PMC5292727/, accessed on 21 January 2022). All PCR reactions were prepared using UV sterilized equipment and negative controls were run alongside the samples. Samples were obtained by amplifying the 16s rRNA V4 region while the ITS were obtained by amplifying the ITS1 region using WineSeq® custom primers (Patent WO2017096385). Samples were pooled in equimolar amounts before sequencing. All the data produced and collected were subsequently analysed through a QIIME-based custom bioinformatics pipeline (Patent WO2017096385, Biome Makers). The first quality control was used to remove adapters and chimeras. After that, the reads were trimmed and Operational Taxonomic Unit (OTU) clusters were performed using a 97% identity. Taxonomy assignment and abundance estimation were obtained comparing OTU clusters obtained with the WineSeq® taxonomy database (Patent WO2017096385). Some of the species were not classified completely at the species level by the DNA sequence but at the genus level. They were defined as X_sp.

### 2.2. Data Analysis

### 2.2.1. Descriptive Analysis

Data from the different samples were put together to build a unique database. A descriptive analysis was carried out with the entire dataset (total known OTUs up to genus level). Total richness was determined as the number of OTUs per kingdom, phylum, class and genus per sample. On the other hand, abundance was expressed as the frequency of occurrence (proportion of sequences) of species *i* at sampling location *j* and was calculated by kingdom, phylum, class and genus. For fungi, richness and abundance were also determined by functional group (symbiont, saprotroph and pathogen) and sampling location. The functional group was classified using the FUNGuild tool [31].

### 2.2.2. Cluster Analysis

A cluster analysis was carried out to analyse the relationships between organisms. In this case, only the OTUs classified at the species level were used in order to know the co-occurrence between pairs of specific organisms. The A*ij* matrix was constructed to contain the frequency of occurrence of species *i* at sampling location *j*, and co-occurrence was measured using three different algorithms for the calculation of species similarity coefficients. The chosen algorithms (Table 1) were the simple correlation or simple matching [32], the Jaccard coefficient [33] and the Russell and Rao coefficient [34]. The algorithms have been widely used in biology for species association studies [35], and are metric co-occurrence coefficients [36].

**Table 1.** Different dissimilarity algorithms used in the study.

| | | Dissimilarity Level Cut | |
|---|---|---|---|
| Coefficient | Reference | Ward | UPGMA |
| Jaccard | Jaccard (1901) | 0.2 | 0.4 |
| Russell & Rao | Russell and Rao (1940) | 0.1 | 0.2 |
| Simple Matching | Sokal and Michener (1958) | 0.2 | 0.2 |

Note: Ward: Ward minimum variance; UPGMA: Unweighted pair group method of averaging.

Cluster analysis results were represented in a dendrogram, according to two criteria. The first one, the Ward minimum variance [37] minimizes the variance within each cluster. The second one, the unweighted pair group method of averaging (UPGMA) [38], defines the distance between two clusters as the arithmetic mean of all the element-to-element distances between the two clusters. The level of dissimilarity at which the dendrogram was cut to obtain the clusters was chosen following a different rule for each method (Table 1) since each method provided quite different distances.

The first correlation matrix showed that the number of species in one sample could misrepresent the final results. For this reason, several tests were carried out in order to obtain the best results. So, the analysis was made (a) with all the data, (b) taking into account only the species that were located in more than one site and (c) weighting the A*ij* matrix by the number of nonzero counts in this species and sample. The co-occurrence of each pair of species was converted to a measure of similarity using statistical methods such as that of Russel and Rao [34], and these pairwise similarities were formed into a dendrogram using hierarchical (agglomerative) clustering. After analysis, the dendrogram created using Russell & Rao and UPGMA was considered the best representation of the relationships of *T. melanosporum* with the other soil organisms.

The Python language (Python Software Foundation) and Spyder IDE (https://www.spyder-ide.org/, accessed on 1 February 2022) were used to calculate the similarities of the different species, together with the packages Numpy (https://numpy.org/, accessed on 1 February 2022), Scipy (https://www.scipy.org, accessed on 1 February 2022), Pandas (https://pandas.pydata.org/, accessed on 1 February 2022) and Fastcluster (https://pypi.org/project/fastcluster/, accessed on 1 February 2022).

## 3. Results

### 3.1. Descriptive Analysis

After discarding singletons, sequence clustering resulted in 5245 OTUs of the total 5271 sequencing reads. From them, 1853 OTUs belonging to the kingdom Fungi and 3392 OTUs to the kingdom Bacteria were identified in the sampled sites. Most of the OTUs were identified down to genus level rather than species level, owing to database limitations. A total of 745 genera were detected across all the samples; 298 genera belonged to fungi and 447 to bacteria. The abundance of the overall 10 most frequent fungi and bacteria species ranged from a minimum of 1% to a maximum of 11% or 4%, respectively (Table 2).

**Table 2.** The 10 most abundant bacterial and fungal species in the studied truffle orchards.

| Fungi | Bacteria |
|---|---|
| *Cryptococcus aerius* (Saito) Nann. (10.9) | *Escherichia* sp. (4.0) |
| *Tuber melanosporum* (8.9) | *Gaiella* sp. (2.4) |
| *Mortierella* sp. (4.1) | *Rubrobacter* sp. (2.3) |
| *Alternaria soliaridae* E.G. Simmons (3.3) | *Novosphingobium* sp. (1.6) |
| *Lewia infectoria* (Fuckel) M.E. Barr & E.G. Simmons (2.8) | *Gemmata* sp. (1.4) |
| *Fusarium* sp. (2.4) | *Skermanella* sp. (1.4) |
| *Tetracladium* sp. (1.6) | *Blastocatella* sp. (1.4) |

**Table 2.** *Cont.*

| Fungi | Bacteria |
| --- | --- |
| *Peyronellaea calorpreferens* (Boerema, Gruyter & Noordel) (1.5) | *Solirubrobacter* sp. (1.3) |
| *Preussia* sp. (1.0) | *Gemmatimonas* sp. (1.2) |
| *Tuber melosporum* (1.0) | *Arthrobacter* sp. (1.1) |

Note: Average relative abundances are given in brackets (%).

### 3.1.1. Fungi

The fungal community across samples was composed of Ascomycota (72% of the total richness), followed by Basidiomycota (22%), Mucoromycota (4%), Olpidiomycota (1%), Zygomycota (<1%), Glomeromycota (<1%) and Chytidriomycota (<1%) (Figure 1).

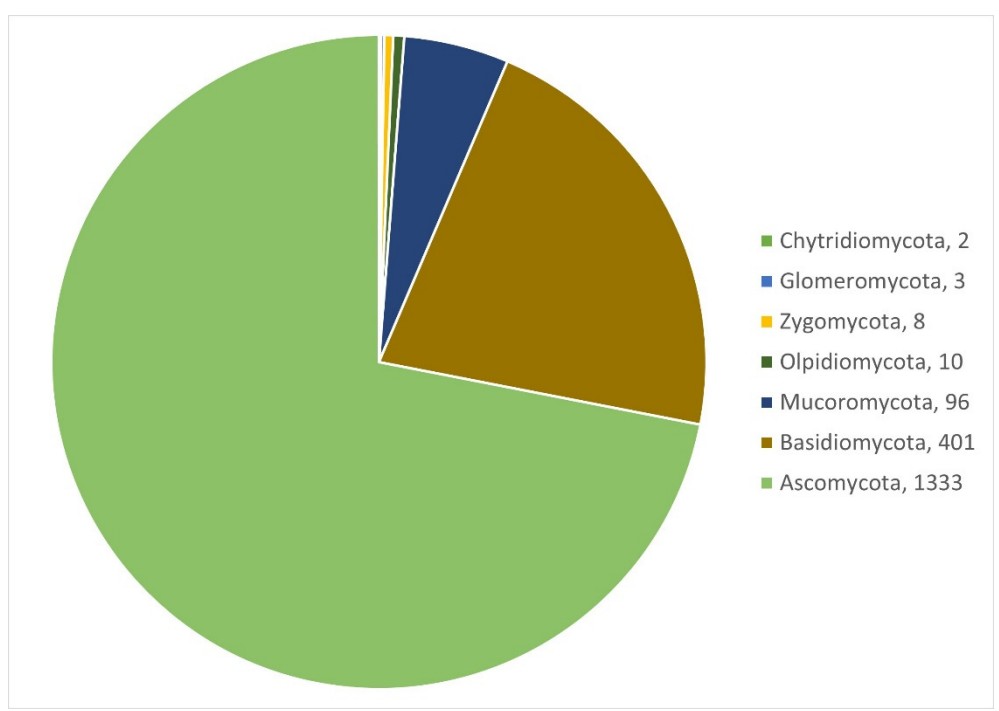

**Figure 1.** Fungal richness, number and proportional distribution of the 1853 fungal OTUs representing all the detected taxonomic phyla.

The five richest taxonomic classes in kingdom Fungi were Sordariomycetes (representing 25% of the richness), followed by Dothiodeomycetes (19%), Agaricomycetes (13%), Eurotiomycetes (11%) and Leotiomycetes (8%).

A total of 298 fungal genera were identified. Those with a greater species richness were genera *Solicoccozyma* (89 OTUs), *Penicillium* (79 OTUs), *Mortierella* (59 OTUs), *Fusarium* (38 OTUs), *Alternaria* (37 OTUs) and *Aspergillus* (37 OTUs). Within functional ecological groups, 162 OTUs were symbionts (9%), 997 saprotrophs (54%) and 694 pathogens (37%).

Regarding the genus *Tuber*, 6 different species were detected: *Tuber aestivum* Vittad., *T. gennadii* (Chatin) Pat., *T. melanosporum*, *T. melosporum (G.Moreno, J.Díez & Manjón) P.Alvarado, G.Moreno, Manjón & J.Díez, T. oligospermum* Tul. & C. Tul. and *T. rufum* Pollini, with *T. melanosporum* being the species with the highest richness and abundance, followed by *T. rufum* and *T. gennadii*. *Tuber melosporum* also showed a high abundance in one of the samples, whilst the rest of *Tuber* spp. were present with a low richness and abundance.

Relative richness was homogeneous in the samples by functional groups (Figure 2a) and by sample (Figure 2b). With respect to the abundance (Figure 2a,c), in several samples, there was a high abundance of *T. melanosporum* and a small abundance of pathogens (samples 3, 5, 6 and 4). In contrast, there was no presence of *T. melanosporum* in sample

number 8, and the abundance of other symbiotic fungi species was minimal with a high presence of pathogenic and saprotrophic species.

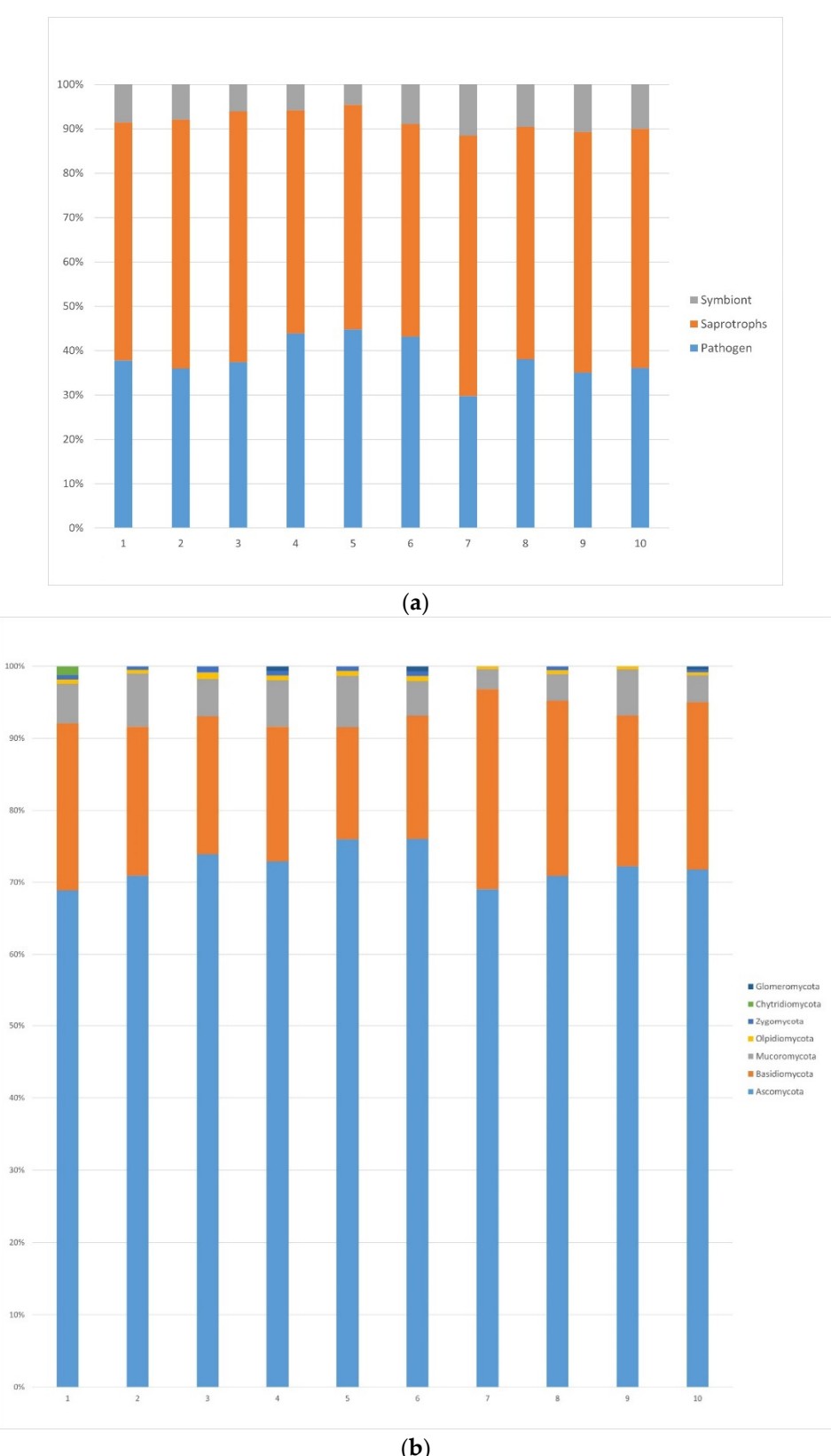

(**a**)

(**b**)

**Figure 2.** *Cont.*

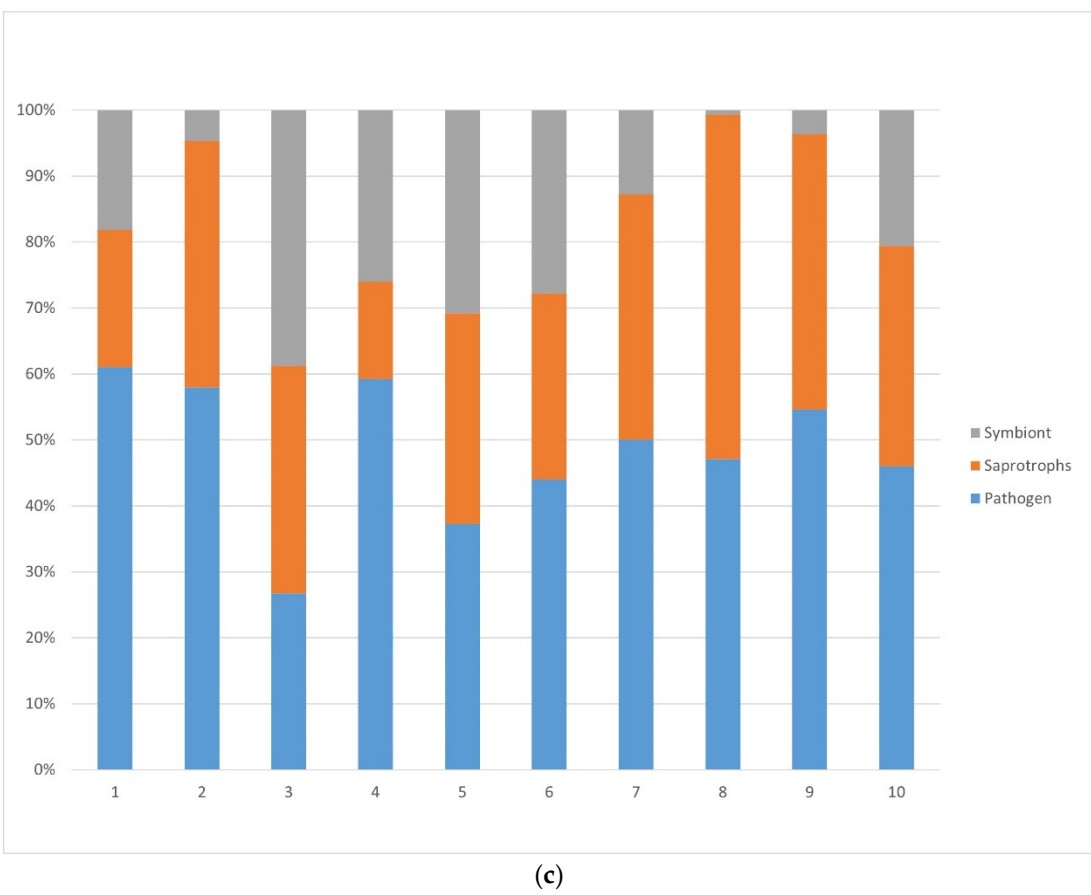

(**c**)

**Figure 2.** (**a**). Fungi ecological groups' richness by sample. (**b**) Fungi phylum richness by sample. (**c**) Ecological groups' abundance by sample.

3.1.2. Bacteria

Regarding bacterial species, the most representative phyla were Proteobacteria (40% of species richness), followed by Actinobacteria (23%), Bacteroidetes (11%) and Firmicutes (9%) (Figure 3a).

Alphaproteobacteria (Figure 3b) was the most species-rich class within Proteobacteria (50%), followed by Gammaproteobacteria (21%), Betaproteobacteria (17%) and Deltaproteobacteria (10%). For Alphaproteobacteria, the most abundant taxonomic orders were Rhizobiales, Rhodospirillales and Sphingomonadales. For Gammaproteobacteria, the most represented taxonomic orders were Xanthomonadales, Enterobacterales and Pseudomonadales. Within Alpha and Gammaproteobacteria, there was a remarkable presence of species well-known by their properties as mycorrhizal helper bacteria, such as the genera *Agrobacterium*, *Azotobacter*, *Burkholderia*, *Bradyrhizobium*, *Enterobacter*, *Pseudomonas* and *Rhizobium*. Figure 3c shows the bacterial richness by sample.

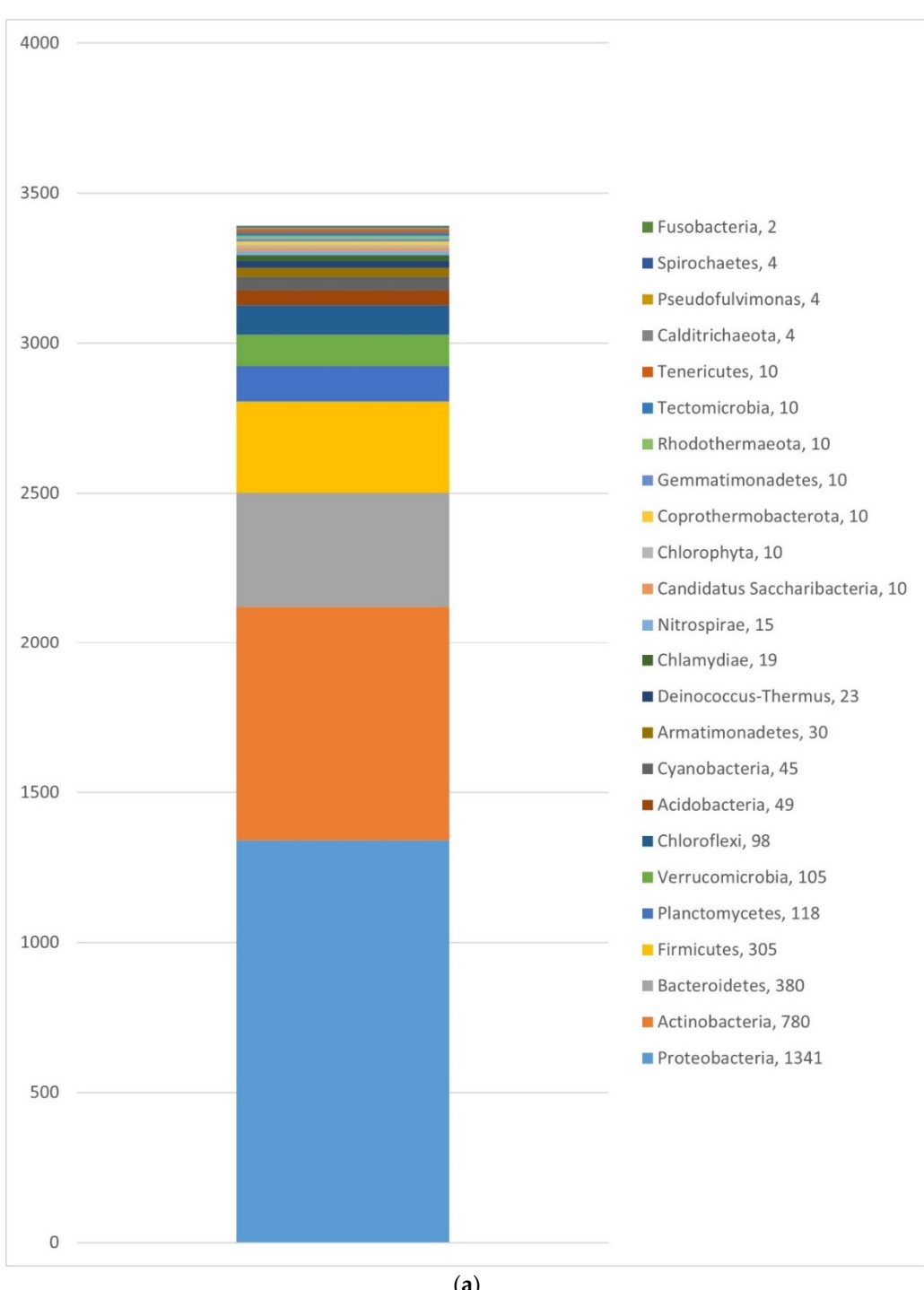

(**a**)

**Figure 3.** *Cont.*

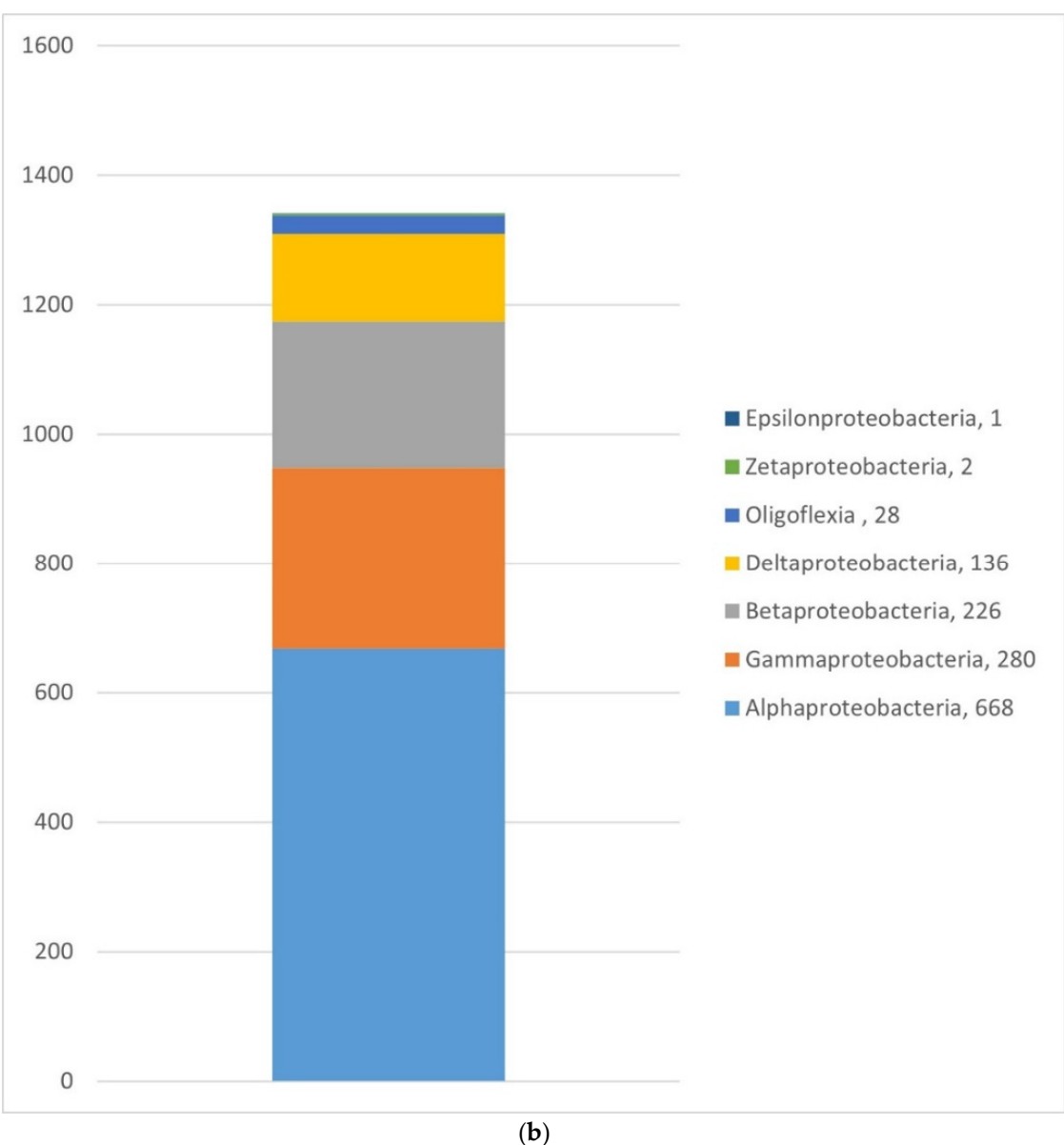

(**b**)

**Figure 3.** *Cont.*

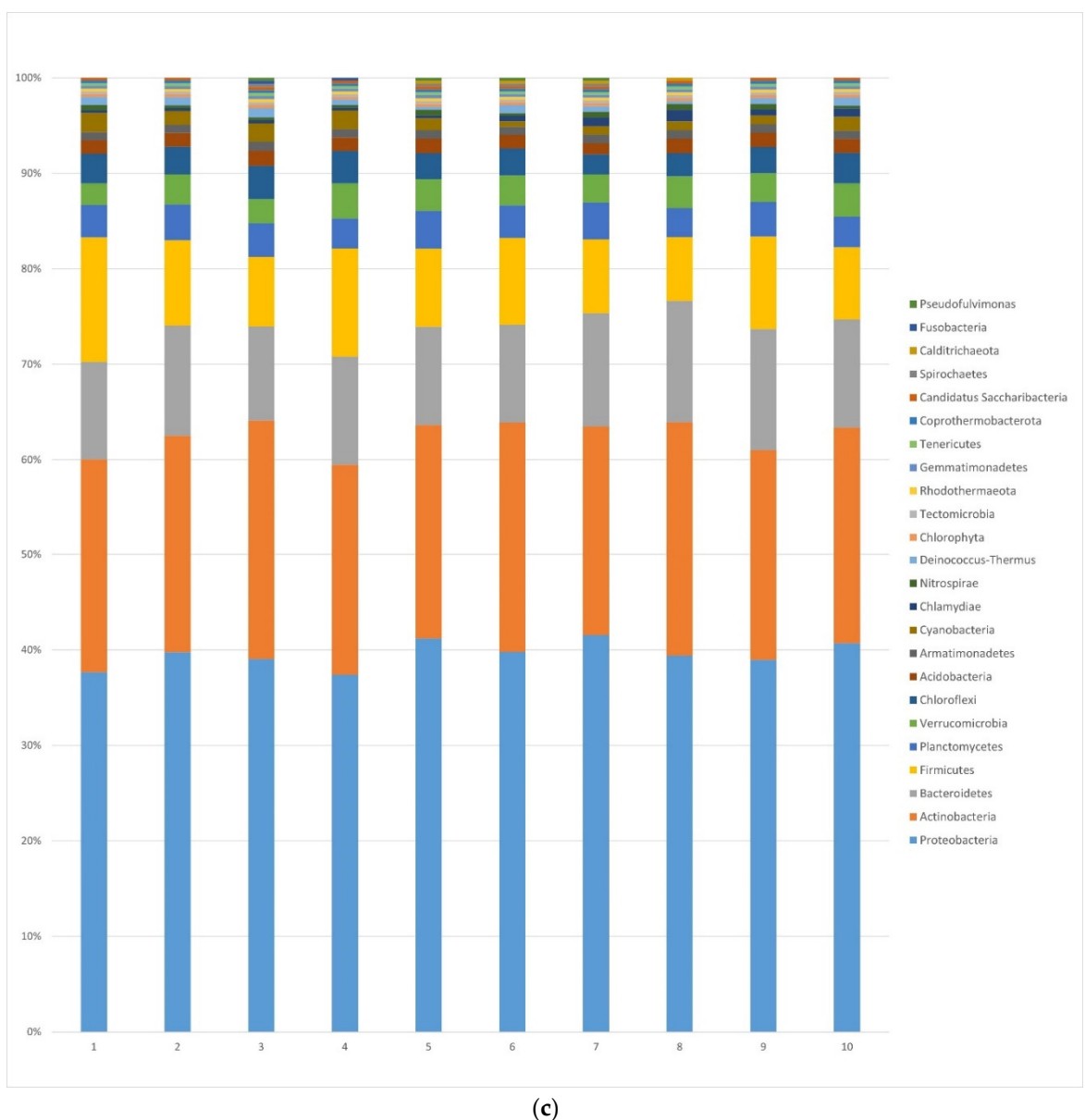

(**c**)

**Figure 3.** (**a**) Bacterial richness, number and proportional distribution of the 3392 bacterial OTUs representing all the detected taxonomic phyla. (**b**) Classes of the phylum Proteobacteria and their number and proportional distribution. (**c**) Bacterial richness by sample.

### 3.2. Dendrogram Analysis

The full dendrogram obtained by the Russel & Rao method and UPGMA is available in Supplementary Figure S1. Figure 4 showed a selected section in the co-occurrence dendrogram to assess the species that appear most frequently with *Tuber melanosporum* in more detail. This selection was carried out because the further the species were removed from the black truffle node in the tree, the more unlikely they would co-occur with *T. melanosporum*.

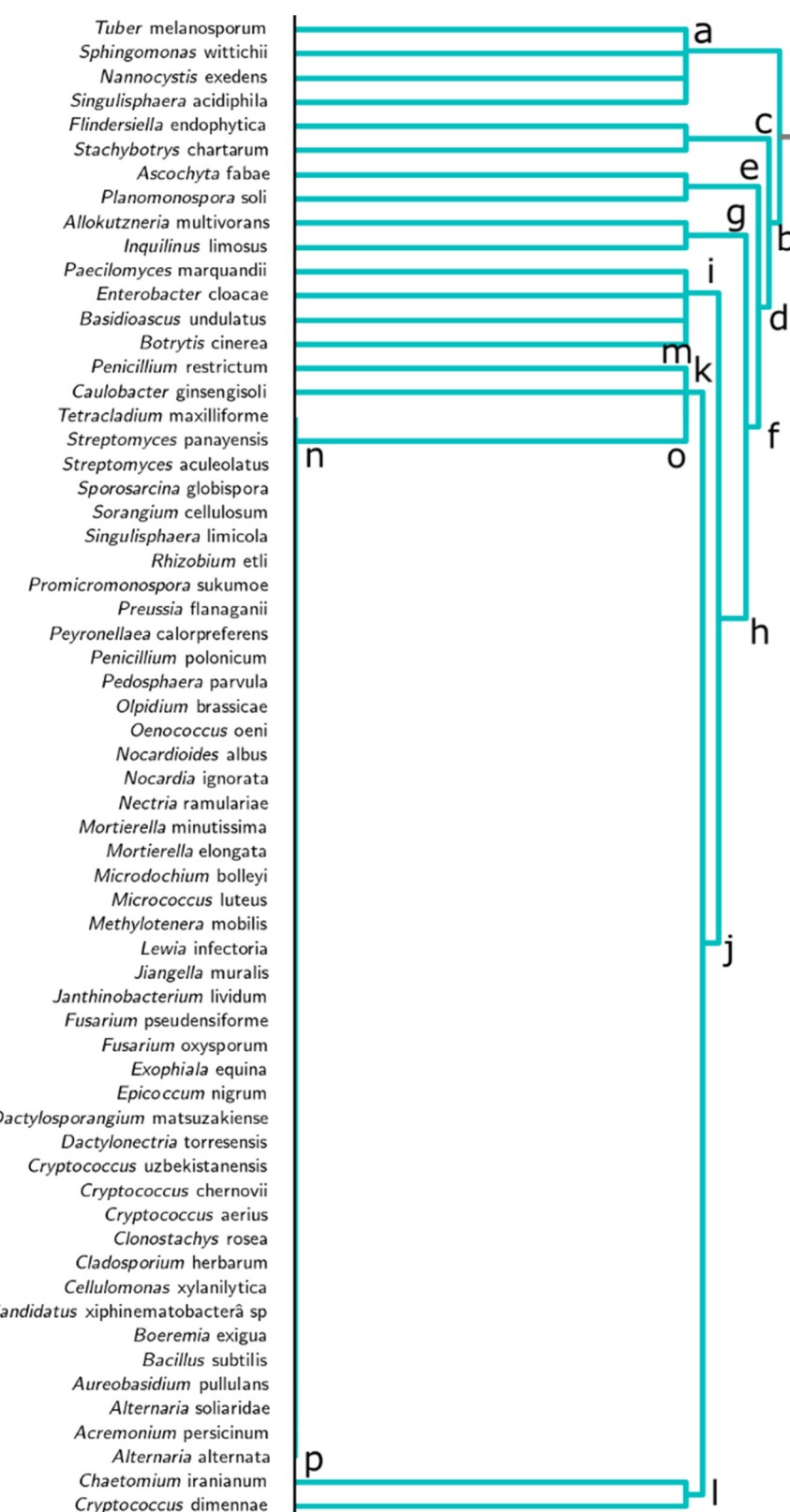

**Figure 4.** Dendrogram of the relationship between *T. melanosporum* and other fungal and bacterial species obtained by Russell & Rao coefficient and UPGMA. N.

The dendrogram can be read as showing which species cluster together at increasing similarity thresholds: at the lowest level (left-hand side of Figure 4), only the most similar

(i.e., most strongly co-occurring) species were clustered. The similarity threshold decreases rightward. Therefore, reading from the right we encounter the most coarse-grained divisions first. So, Figure 4 shows a division between two main branches (*a*) and (*b*), where each letter name refers to the sub-tree rooted at that point. The first subtree (*a*) was composed of *T. melanosporum* and the bacterial species *Sphingomonas wittichii* Yabuuchi, *Nannocystis exedens* Reichenbach. and *Singulisphaera acidiphila* Kulichevskaya. The other subtree (*b*) was composed of species from *Flindersiella endophytica* Kaewkla & Franco (bacteria species) to *Cryptococcus dimennae* Fell & Phaff (fungal species). In the subtrees under subtree (*b*), bacterial species were mixed together with fungal species. In the subtrees closer to *T. melanosporum*, there were more bacterial species than fungal species, such as in the subtree (*g*), where there was no presence of fungal species. However, the more distant black truffle was, the more abundant fungal species were, such as in subtree (*i*), composed of three fungal species and one bacterial species, and the subtree (*p*), with 26 fungi species and 18 bacterial species.

In the subtrees of subtree (*b*), within bacterial species, there were different Gram-negative species, which are mainly pathogenic, such as *Inquilinus limosus* Coenye., *Janthinobacterium lividum* Eisenberg, *Methylotenera mobilis* Kalyuzhnaya. or the myxobacteria *Sorangium cellulosum* (Brockman) Reichenbach. Another bacterial species found was *Rhizobium etli* Segovia. Gram-positive species were also found, such as *Bacillus subtilis* (Ehrenberg) Cohn., *Cellulomonas xylanilytica* Rivas., *Micrococcus luteus* (Schroeter) Cohn., *Nocardioides albus* Prauser., *Oenococcus oeni* (Garvie) Dicks. *Streptomyces panayensis* Hasegawa (which produce exotoxicity, i.e., antibiotics to kill other individuals), and *Sporosarcina globispora* (Larkin and Stokes) Yoon.

Regarding fungal species, subtrees of tree (*b*) showed many fungal saprotrophic species, such as *Tetracladium maxilliforme* (Rostr.) Ingold and *Paecilomyces marquandii* (Massee) S. Hughes, and pathogenic species such as *Boeremia exigua* (Desm.) Aveskamp, Gruyter & Verkley, *Ascochyta fabae* Speg, *Alternaria alternata* (Fr.) Keissl (one of the most important pathogens belonging to the Ascomycota phylum), *Peyronellaea calorpreferens* Boerema, Gruyter & Noordel.) Aveskamp, Gruyter & Verkley, *Botrytis cinerea* Pers. and its antagonist *Penicillium polonicum* K.W. Zaleski, *Cryptococcus* uzbekistanensis Á. Fonseca, Scorzetti & Fell., *C. chernovii* Á. Fonseca, Scorzetti & Fell or *C. aerius*, *Olpidium brassicae* (Woronin) P.A. Dang., *Dactylonectria torresensis* (A. Cabral, Rego & Crous) L. Lombard & Crous, *Fusarium pseudensiforme* Samuels, Nalim & Geiser and *F. oxysporum* Schltdl. Endophitic species such as *Aureobasidium pullulans* (de Bary & Löwenthal) G. Arnaud, *Clonostachys rosea* (Link) Schroers, Samuels, Seifert & W. Gams (which protects against *Botrytis cinerea*), *Exophiala equina* (Pollacci) de Hoog, V.A. Vicente, Najafz., Harrak, Badali & Seyedm or *Epicoccum nigrum* Link. were also associated with *T. melanosporum* in the dendrogram.

## 4. Discussion

Our results provide a comprehensive description of the microbiomes found in truffle orchards in Northern Spain, as well as the fungal and bacterial species most significantly associated with *T. melanosporum* mycelium in the soil. It is essential to understand the functioning of the microbial communities and their interactions in order to acquire adequate ecological knowledge, preserve and utilize biodiversity, and thus improve truffle plantations. This analysis advances the understanding of how the microbial community is composed and how interspecific relationships and interactions drive the dynamics in truffle plantation soils.

Our results showed that regarding the fungal community, Ascomycota was the phylum with the highest richness, and within it, *T. melanosporum* was the most abundant species. Previous researchers also found that Ascomycota was the most abundant phylum in black truffle plantations [1,39].

Our findings showed that after *T. melanosporum*, the most abundant species were *Mortierella* sp. *Mortierella* is a genus of saprotrophic fungi of the phylum Mucoromycota, promoting endophytic fungi, which produce indole acetic acid (IAA) and modulate phy-

tohormone levels in plant roots. The presence of *Mortierella* spp., and other endophytic or plant pathogenic fungi, can affect colonization of plant roots or fructification of ectomycorrhizal fungi [39,40]. The genus *Mortierella* has been observed in previous truffle studies [41] and associated with *Tuber magnatum* Picco's most productive sites [40]. It has also been reported to assist mycorrhizal fungi in phosphorus acquisition [42,43]. The species *Mortierella minutissima* Tiegh and *M. elongata* Linnem were found to be associated with black truffle in our dendrogram analysis.

Among the most abundant genera, there was also *Cryptococcus* of the phylum Basidiomycota. Eighteen species of the genus *Cryptococcus* were detected in our study, such as the species *Cryptococcus uzbekistanensis*, *C. chernovii* and *C. aerius* which were also found to be associated with black truffle in the dendrogram. Zacchi et al. [44] suggested a certain degree of specificity between different *Cryptococcus* spp. (*Cryptococcus albidus* (Saito) C.E. Skinner and *C. humicola* (Dasz.) Golubev.) and *Tuber* ecosystem, indicating an active role of these microorganisms in fungal spore dispersion and/or germination in soil. Moreover, Zacchi et al. [44] showed that the pectinolytic capacity of *C. albidus* could indicate a helper role for the ectomycorrhizal occupation of the plant root system.

Considering the composition of the different trophic groups of the fungal communities, it was observed that the greater abundance of symbiont fungi was due exclusively to the high presence of truffle in these samples (3, 5 and 6), given that these plots did not have a higher richness of other mycorrhizal species (Figure 2a,c). In a recent study, Oliach et al. [5] found no association between the abundance of *T. melanosporum* and the presence of other ectomycorrhizal fungi. Furthermore, in those sites with a higher abundance of *T. melanosporum*, the abundance of pathogenic organisms was lower. This result also agreed with that found by Oliach et al. [5], who reported a negative correlation between the abundance of *T. melanosporum* and the abundance of putative plant pathogens.

Concerning the bacterial community, Proteobacteria and Actinobacteria were the most abundant phyla, which agreed with the findings reported in previous studies. Antony-Babu et al. [45] showed similar results in *T. melanosporum* plantations and Mediavilla et al. [22] found that Proteobacteria (25% of the total bacterial richness) and Actinobacteria (14%) were the phyla most abundant in *Boletus edulis* Bull. productive forests. As well as this, Zhang et al. [46] found that Proteobacteria (52%) and Actinobacteria (24%) were also the most abundant phyla in *Quercus acutissima* Carruth. forests, where microbial communities of *T. indicum* were studied. A high level of Alphaproteobacteria biodiversity was found in *T. pseudoexcavatum* Y. Wang, G. Moreno, Riousset, Manjón & G. Riousset, *T.* sinoaestivum J.P. Zhang & P.G. Liu, *T. indicum* Cooke & Massee and *T. panzhihuanense* X.J. Deng & Y. Wang plantations in China [41,47]. Alphaproteobacteria has been considered as a class with a broad adaption to many habitats and ecosystems [48] and as a dominant class in *Tuber* plantations [45,49], which could suggest an adaptive or competitive mechanism with other bacterial phyla [50].

Deveau et al. [51] found an enrichment in Alpha and Gammaproteobacteria in *T. melanosporum* plantations, which appeared to indicate their functional importance [23]. In our study, for Alphaproteobacteria the most abundant orders were Rhizobiales, Rhodospirillales and Sphingomonadales and for Gammaproteobacteria the most represented orders were Xanthomonadales, Enterobacterales and Pseudomonadales. Within these orders, the presence of species of the genera *Agrobacterium*, *Azotobacter*, *Burkholderia*, *Bradyrhizobium*, *Enterobacter*, *Pseudomonas* and *Rhizobium* was observed. All of these are well-known for their properties as mycorrhizal helper bacteria (MHB) [20], which play a key role in the establishment of plant–fungal symbioses [52]. The mechanisms by which bacteria interact with fungi involve the stimulation of mycelial growth, the increase of contact points between roots and fungi and the reduction of environmental stress on the mycelium [53,54]. These bacteria could also inhibit colonization by pathogen or antagonist fungi [20]. Mamoun and Olivier [55] stated an indirect helper effect of soil pseudomonads on the *T. melanosporum* symbiosis, and Dominguez et al. [56] reported a significant increase in colonization of *Pinus halepensis* Mill. roots by *T. melanosporum* in the presence of *Pseudomonas fluorescens* Migula.

The presence of *Bradyrhizobium* spp. was also highlighted by Barbieri et al. [57] in a *T. magnatum* study.

*Tuber melanosporum* is thought to modify and affect selection in the associated microbiome. Our dendrogram results, which showed additional information about the interactions of fungi and bacteria with *T. melanosporum*, could support this assumption. The dendrogram showed that *T. melanosporum* appeared alone in the first group without the presence of other fungal species. It could indicate the competitive effect of *T. melanosporum* with other ectomycorrhizal fungi. Mello et al. [1] found that in the places where truffles were the dominant fungus, the presence of Basidiomycota ectomycorrhizal fungi decreased. Streiblová et al. [58] suggested that truffles could adopt an efficient survival strategy by spreading their metabolites, which are regarded as having allelopathic effects on the herbaceous plants and the microorganisms in the rhizosphere, forming a non-vegetated area around the host plant, known as a brûlé [1]. Mello et al. [59] studied the composition of fungal communities in *T. melanosporum* orchards and found lower fungal biodiversity inside the brûlé. Furthermore, Napoli et al. [60] reported a competitive effect of *T. melanosporum* on other ectomycorrhizal species within the brûlé. That suggests that the truffle has an impact on certain mycorrhizal fungi [60]. Furthermore, at the bacteria species level, some phyla were found more frequently in the brûlé, such as Firmicutes (e.g., *Bacillus*), several genera of Actinobacteria and a few Cyanobacteria. These findings corroborate that not only fungal communities are affected by *T. melanosporum*, but also other microorganisms.

Our findings showed that *T. melanosporum* tended to closely co-occur with different bacterial species in the first cluster. So, *Sphingomonas wittichii*, *Nannocystis exedens* and *Singulisphaera acidiphila* were bacterial species found to closely co-occur with *T. melanosporum*. *Sphingomonas* spp. are widely distributed in the root rhizosphere [61]. They were found in *T. magnatum* communities studied by Barbieri et al. [18]. *Nannocystis exedens* is a micropredator that obtains its nutrients by lysing and killing cells of other bacteria and yeasts [62]. This bacterial species generates some volatile compounds which could cause inhibition of plant pathogens [63]. Some of the volatile compounds found in *Nannocystis* spp. (Phenylethanol and Dimethyl-(methylethyl pyrazine)) were also found in *T. melanosporum* [64]. It has been reported that interactions via volatile compounds also occur in the soil community [65].

The bacterial species *Streptomyces panayensis* was observed in the dendrogram associated with black truffle. This species plays an important role in plant growth, helping to improve shoot and root growth, biological nitrogen fixation and the solubilisation of minerals [66]. Tarkka et al. [67] stated the benefit of *Streptomyces* spp. as helper bacteria fostering nutrient acquisition and enhancing the formation of mycorrhizas in filamentous fungi. The bacterial species *Rhizobium etli* was also observed in the dendrogram. *Rhizobium* is a bacterial genus that has the ability to fix nitrogen and help in increasing plant growth through enriching nutrients, solubilizing phosphates and producing phytohormones [68]. The results obtained suggest a beneficial role of some bacteria present in the mycorrhizosphere of *T. melanosporum*.

The dendrogram showed also some fungal species, such as *Stachybotrys chartarum* (Ehrenb.) S. Hughes and *Ascochyta fabae*, close to black truffle (subtree *bc*). These fungal species are plant pathogens found in the rhizosphere of different ecosystems [69,70]. Furthermore, the fungus *Paecilomyces marquandii* (a saprotrophic species) was observed in the dendrogram. This species has been observed in *Tuber brumale* Vittad., *T. borchii* Vittad. and *T. melanosporum* studies [71,72].

The fungus *Tetracladium maxilliforme* was also observed in the dendrogram. Zambonelli et al. [73] showed the capacity of this species to affect the mycorrhization process and it was reported by Pacioni et al. [72] as a truffle-inhabiting fungus. This fungal species seems to play an important role in the formation and maturation of the *T. melanosporum* ascoma and in the production of the volatile organic compounds, which are responsible for the aroma of the truffle.

Our results revealed *Fusarium* sp. as one of the most abundant species (Table 2). It also appeared in the dendrogram associated with *T. melanosporum*. *Fusarium* is a genus of saprotrophic filamentous fungi which are facultative pathogens of plants belonging to the phylum Ascomycota [74]. Oliach et al. [75] found two species of this genus as indicator species in the samples with the highest mycelium content of *T. melanosporum*. These authors related this greater abundance of *Fusarium* in soils with *T. melanosporum* to the presence of plants weakened during the formation of the brûlé, which are more susceptible to being attacked by the pathogen. Furthermore, Liu et al. [76] found a higher amount of *Fusarium* spp. in the soils of young plantations of *T. melanosporum*, which decreased with increasing age of the stand. Due to the fact that the plantations in which we have collected the samples were generally young (5 to 12 years old), the presence of *Fusarium* in the soil agrees with what was found by Liu et al. [76]. Despite this, these putative plant pathogens did not appear to harm the trees in the plantations, since Oliach et al. [75] also found a greater development of the root collar in these soils with a higher abundance of *T. melanosporum* and pathogens.

The emerging high-throughput sequencing technology allows us to explore new approaches to study the complex community structure of soil fungi in different ecosystems. Our study, based on biotic interaction networks could detect changes in microbial communities, providing simple and operational indicators of ecosystem quality and functioning [30].

Our results reveal that *Tuber melanosporum* co-occurs with different bacterial and fungal species. Co-occurrence analyses have been applied to different plant root systems, and they allow us to infer how diverse are the taxonomic/functional groups [77], and how those networks are compartmentalized [78].

Soil communities can be quite diverse in their composition due to the soil characteristics, the host tree species [79], the fungal symbionts [80] and the composition of the soil bacterial communities. Thanks to the dendrogram analysis it was possible to determine the hierarchical clustering of the different fungal and bacterial species with respect to *T. melanosporum* and look into their possible roles with respect to it, and even to know species as potential bioinoculants, agronomical amendments to increase mushroom productivity through growth promotion or as biocontrol agents to control pests.

## 5. Conclusions

Identifying and understanding the ecology and role of the microbiome in *T. melanosporum* plantations is crucial to preserving these fungal ecosystems and performing adequate management. Our results add new knowledge on the associations by which fungi and bacteria jointly act as drivers of the *T. melanosporum* microbiome. Thanks to metagenomic and dendrogram analysis, it was possible to identify some relevant fungal and bacterial species occurring in truffle plantations such as *Sphingomonas wittichii*, *Nannocystis exedens* and *Singulisphaera acidiphila*. Thus, this study provides a better understanding of the *T. melanosporum* × *Q. ilex* systems. Despite these findings, there is still much uncertainty about the association between individual bacterial and fungal species on the development of truffles and so further research is needed in this field.

**Supplementary Materials:** The following supporting information can be downloaded at: https://www.mdpi.com/article/10.3390/f13030385/s1, Figure S1: Dendrogram obtained by the Russel and Rao method and UPGMA.

**Author Contributions:** Conceptualization, C.H.d.A., J.M., J.O. and O.M.; methodology, C.H.d.A., J.M., S.M., M.H.-R. and O.M.; data curation and formal analysis, C.H.d.A., S.A., S.M. and J.M.; writing—original draft preparation, C.H.d.A. and O.M.; writing—review and editing, C.H.d.A., S.A. and O.M.; supervision, C.H.d.A., J.O., M.H.-R. and O.M. All authors have read and agreed to the published version of the manuscript.

**Funding:** This study was possible through the funding from the Project 'CLU-2019-01-iuFOR Institute Unit of Excellence' of the University of Valladolid, funded by the Junta de Castilla and co-financed by the European Union (ERDF "Europe drives our growth"), the European Union's Horizon 2020 research and innovation programme under grant agreement No 734907 and the Research Project 04/16/PA/0001 (Junta de Castilla y León).

**Acknowledgments:** We want to express gratitude to Esther de Andres, Esther del Amo and Ana Martinez due to their help in looking for information about the species, Iván Franco for the help with samplings and Luis Santos for data collection and manuscript review. In addition, the authors would also thank the MSCA-RISE "3D-NEONET: Drug Discovery and Delivery NEtwork for ONcology and Eye Therapeutics" project for the opportunity of working close to the UCD in the development and use of new programming techniques applied to data analysis.

**Conflicts of Interest:** The authors declare no conflict of interest.

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
