# Peer review of "Fungal and Bacterial Communities in Tuber melanosporum Plantations from Northern Spain"

_forests, doi:10.3390/f13030385_

Round 1
Reviewer 1 Report
Some general comments are on the annotated manuscript. These have to do with grammar and usage.
I find the colors difficult to separate on Figure 1
It might be good to consider defining pathogen. You are largely mentioning plant pathogens. There might also be a discussion of mycoparasitism.

Author Response
Manuscript title: Fungal and bacterial communities in Tuber melanosporum plantations from Northern Spain
Authors: Celia Herrero de Aza, Dr., Sergio Armenteros, PhD.; James McDermott, Dr., Stefano Mauceri, Dr., Jaime Olaizola, Dr., María Hernández-Rodríguez, Dr., Olaya Mediavilla, Dr.
Response to Referee 1
Dear reviewer, first of all, we would like to thank you for the comments. We appreciate your positive observations and deep concern towards our work. By replying to your specific comments below we hope to clarify the doubts raised in the first version of our manuscript.
Comment 1. This phrase is unnecessary
Response. The sentence was modified with referee 2 suggestion and now it is clearer.
Comment 2. Hypogeous
Response. The mistake was corrected.
Comment 3. here and throughout. Pathogenic is mentioned it should be pointed out that these are pathogens of plants. There may also be an issue of mycoparasitism that is not mentioned anywhere but could have an effect on the outcome depending on the fungi found.
Response. According with the reviewer comment, it has been clarified in the text that we are speaking of plant pathogens.
Comment 4. here an throughout it would be good to avoid related which seems to imply a taxonomic affinity.
Response. The document was revised and related was changed by following synonyms associated with or linked to.
Comment 5. spell the first word of a sentence
Response. The suggestion was carried out.
Comment 6. one sentence paragraphs should be avoided
Response. The sentence was included in the following paragraph.
Comment 7. Ectomycorrhizal fungi
Response. The mistake was corrected.
Comment 8. unclear what "them" refers to
Response. Them was changed by bacterial species.
Comment 9. Bacterial
Response. The mistake was corrected.
Comment 10. does this need to be defined?
Response. A definition was added at this point: area devoid of vegetation around the symbiotic plants, where the fruiting bodies of T. melanosporum are usually collected.
Comment 11. rephrase here to avoid repeating the information
Response. The paragraph was rewritten to avoid repeated information.
Comment 12. modify here. This implies that Mortierella is an Ascomycota.
Response. The previous sentence: Within the phylum Ascomycota, it is remarkable that the most abundant species, after T. melanosporum, was Mortierella sp
Was changed by
Our findings showed that after T. melanosporum, the most abundant species was Mortierella sp.
Comment 13. spp. since there are several
Response. The mistake was corrected.
Comment 14. modify here. Are you talking about one or several?
Response. The sentence was clarified.
Comment 15. genera not genus
Response. The mistake was corrected.
Comment 16. this repeats
Response. The paragraph was rewritten.
Comment 17. not sure this is clear.
Response. The paragraph was revised and some modifications have been done in order to a better understanding. The final paragraph was “Considering the composition of the different trophic groups of the fungal communities, it was observed that the greater abundance of symbiont fungi was due exclusively to the high presence of truffle in these samples (3, 5 and 6), given that these plots did not have a higher richness of other mycorrhizal species (Figure 2a,c)”.
Comment 18. rephrase here.
Response. It has been rephrased for a better understanding. “In our study, for Alphaproteobacteria the most abundant orders were Rhizobiales, Rhodospirillales and Sphingomonadales and for Gamma-proteobacteria the most represented orders were Xanthomonadales, Enterobacterales and Pseudomonadales”.
Comment 19. Within them, it was observed the presence of species of the genera…
Response. It was changed for “Within these orders” for a better understanding.
Comment 19. spell out
Response. Your suggestion was carried out.
Comment 20. rephrase here
Mello et al. [1] found that when truffles were the dominant fungus, the presence of Basidiomycota ectomycorrhizal fungi decreased
Response. It has been changed to “Mello et al. [1] found that in the places where truffles were the dominant fungus, the presence of Basidiomycota ectomycorrhizal fungi decreased”.
Comment 21. delete sp. you are talking about the genus. Otherwise write it out "Species of Rhizobium"
Response. Your suggestion was carried out and the term sp was deleted.
Comment 22. co-occurs
Response. The mistake was corrected.
Summary: As a result of the revision, the errors and suggestions marked in the pdf were carried out. As well as this, the introduction and discussion were improved. All the comments and suggestions received were highlighted and all the modifications built a new version of the manuscript. Any changes related to The English revision can be supervised and easily viewed by the editors and reviewers with “Track Changes” function”.
Reviewer 2 Report
Dear authors & editors,
Here is the review of the paper titled "Fungal and bacterial communities in Tuber melanosporum plantations from Northern Spain" written by Celia Herrero de Aza & co-authors.
The aim of the paper was to research fungal and bacterial communities associated with commercially important truffle species Tuber melanosporum in 10 holm oak plantations in Northern Spain. The most abundant taxa associated with T. melanosporum were bacteria Singulisphaera limicola, Nannocistis excedens and Sporosarcina globispora; and fungi Mortierella elongata, M. minnutissima, Cryptococcus uzbekistanensis, C. chernovii and C. aerius.
It seems that the authors incorrectly use abbreviations sp. and spp. So the reader can misunderstand the results obtained in the study.
E.g. Fusarium sp. (singular) means Fusarium species (one species) that is not identified to the species level but to the genus level. Also, the authors frequently use (e.g.) genus Rhizobium sp. If you talk about the genus it is unnecessary to have sp. after the genus name.
Fusarium spp. (plural) means 2 or more Fusarium species (whether identified to the species or genus level).
In table 2. What do you mean with (e.g.) Fusarium sp. (one or more Fusarium species?) Please go through the whole manuscript text and change sp. and spp. accordingly.
The English language used needs to be improved because some sentences are unclear and ambiguous. Methods should be better explained. Other than that, the paper contains many errors that are marked in the pdf of the manuscript attached. Additionally, some conclusions are overemphasized (e.g.) "exhaustive analysis of the diversity, structure, and composition of fungal and bacterial communities". I am not sure whether it is appropriate to characterize the study as exhaustive since the authors do not list all fungal and bacterial taxa found in plantations. OTU table is not included. In the dendrogram, all taxa are identified to the species level while on the other hand, the authors write that the species are mostly identified to the genus level. Please explain that in the text.
My recommendation is a major revision.
Best, reviewer

Author Response
Manuscript title: Fungal and bacterial communities in Tuber melanosporum plantations from Northern Spain
Authors: Celia Herrero de Aza, Dr., Sergio Armenteros, PhD.; James McDermott, Dr., Stefano Mauceri, Dr., Jaime Olaizola, Dr., María Hernández-Rodríguez, Dr., Olaya Mediavilla, Dr.
Response to Referee 2
Dear Reviewer, first of all, we would like to thank you for the comments and suggestions, contributing to improve the quality of the document. Following your suggestions, further in-depth revision and correction has been made on the whole document, mainly with respect to different terms and the use of the abbreviations sp. and spp. In addition, an English revision was carried out.
Comment 1. Delete "belonging to the Tuber genus".
Response. Your suggestion was carried out.
Comment 2. hypogeum -> hypogeous
Response. The mistake was corrected.
Comment 3. fungi -> fungal
Response. The mistake was corrected.
Comment 4. minnutissima -> minutissima
Response. The mistake was corrected.
Comment 5. delete comma
Response. Your suggestion was carried out.
Comment 6. T. -> Tuber
Response. Your suggestion was carried out.
Comment 7. bacteria -> bacterial
Response. The mistake was corrected.
Comment 8. truffles -> truffle
Response. The mistake was corrected.
Comment 9. In which time of the year you collected the samples?
Response. The month was added to the end of the sentence.
Comment 10. m.a.s.l. -> m a.s.l.
Response. The mistake was corrected.
Comment 11. Which is the sampling distance between each of 4 cores used to obtain composite sample?
Response. The information was added at the end of the sentence.
Comment 12. Quiagen -> Qiagen
Response. The mistake was corrected.
Comment 12. Reference? Or describe the protocol.
Response. A reference was added.
Comment 13. assignation -> assignment
Response. The mistake was corrected.
Comment 14. saprophyte -> saprotroph
Response. The mistake was corrected.
Comment 15. Numpy -> Numpy (https://numpy.org/)
Response. The information was added.
Comment 16. In dendrogram all taxa are identified to the species level. Please explain.
Response. The referee is right. For the descriptive analysis the entire dataset was used (OTUs with genus level) because we needed to know which is the abundance and richness at different levels (phylum, genus…etc). However, for the cluster analysis only the species was used because we needed to know the co-ocurrence between organisms. If we do not know which specific organism is X_spp in site A, we could not know if it is the same organisms in site B. To make it clearer, two sentences were added in the material and method section in order to explain properly which data were used in each case, as follows:
2.2. Data analysis
Data from the different samples were put together to build a unique database. A descriptive analysis was carried out with the entire dataset (total known OTUs up to genus level).
Cluster analysis
A cluster analysis was carried out to analyse the relationships between organisms. In this case only the OTUs classified at species level were used in order to know the co-occurrence between pairs of specific organisms.
Comment 17. put all "sp." in regular font, not italic
Response. The mistake was corrected along the entire document.
Comment 18. delete all "spp." in this sentence
Response. The mistake was corrected.
Comment 19. saprophytes -> saprotrophs
Response. The mistake was corrected.
Comment 20. Tuber spp. genus -> genus Tuber
Response. Your suggestion was carried out.
Comment 21. put all "Tuber" -> "T." in this sentence
Response. Your suggestion was carried out.
Comment 22. T. -> Tuber
Response. Your suggestion was carried out.
Comment 23. pathogen -> pathogenic
Response. Your suggestion was carried out.
Comment 23. saprophyte -> saprotrophic
Response. Your suggestion was carried out.
Comment 24. saprophytes -> saprotrophs
Response. The mistake was corrected in the figure.
Comment 25. saprophytes -> saprotrophs
Response. The mistake was corrected in the figure.
Comment 26. bacteria -> bacterial
Response. The mistake was corrected.
Comment 27. Figura -> Figure
Response. The mistake was corrected.
Comment 28. delete "spp." from this sentence
Response. The mistake was corrected.
Comment 29. Tuber -> T.
Response. Your suggestion was carried out.
Comment 30. figure -> Figure
Response. Your suggestion was carried out.
Comment 31. bacteria -> bacterial
Response. The mistake was corrected.
Comment 32. fungus -> fungal
Response. The mistake was corrected in the entire paragraph.
Comment 33. bacteria -> bacterial
Response. The mistake was corrected in the entire paragraph.
Comment 34. pathogen species -> pathogenic
Response. The mistake was corrected.
Comment 35. Respect to fungi species -> Regarding fungal species
Response. Your suggestion was carried out.
Comment 36. fungus saprophic ->fungal saprotrophic
Response. The mistake was corrected.
Comment 37. pathogen -> pathogenic
Response. The mistake was corrected.
Comment 38. Cryptococcus -> C.
Response. Your suggestion was carried out.
Comment 39. protect -> protects
Response. The mistake was corrected.
Comment 40. delete "sp."
Response. Your suggestion was carried out.
Comment 41. saprophytic -> saprotrophic
Response. The mistake was corrected.
Comment 42. sp. -> spp.
Response. The mistake was corrected.
Comment 43. by -> of
Response. The mistake was corrected.
Comment 44. sp.? This means that one species (= singular) of the genus Mortierella is found but it is not identified to the species level. Spp. (= plural) means that more than one species of the genus is found. What is the truth?
Response. The confusion was resolved.
Comment 45. Mortierella -> M.
Response. Your suggestion was carried out.
Comment 46. genus -> genera
Response. The mistake was corrected.
Comment 47. delete "spp."
Response. The mistake was corrected in that place and in the next one.
Comment 48. Belonged -> Belonging
Response. That paragraph was rewritten with one comment from reviewer 1. The term belonged was dropped.
Comment 49. sp. -> spp.
Response. The mistake was corrected.
Comment 50. Cryptococcus -> C.
Response. Your suggestion was carried out.
Comment 51. bacteria -> bacterial
Response. The mistake was corrected.
Comment 52. Tuber -> T.
Response. Your suggestion was carried out in that and the following three ones.
Comment 53. bacteria -> bacterial
Response. The mistake was corrected.
Comment 54. delete" [24]."
Response. The mistake was corrected.
Comment 55. the orders most abundant were -> the most abundant orders were
Response. The mistake was corrected.
Comment 56. delete "sp." from all places in this sentence
Response. Your suggestion was carried out.
Comment 57. them -> these
Response. Your suggestion was carried out.
Comment 58. Tuber -> T.
Response. Your suggestion was carried out.
Comment 59. T. -> Tuber
Response. Your suggestion was carried out.
Comment 60. bacteria -> bacterial
Response. The mistake was corrected in that place and the next one.
Comment 61. sp. -> spp.
Response. The mistake was corrected.
Comment 62. bacteria species -> bacterial species
Response. The mistake was corrected.
Comment 63. sp. -> spp.
Response. The mistake was corrected in that place and the next one.
Comment 64. delete "sp."
Response. The mistake was corrected.
Comment 65. closed -> close ?
Response. The mistake was corrected.
Comment 66. saprophitic -> saprotrophic
Response. The mistake was corrected.
Comment 67. unclear phrase?
Response. The phrase was rewritten.
Comment 68. what do you mean by isolations?
Response. The phrase was rewritten.
Comment 69. fungi -> fungus
Response. The mistake was corrected.
Comment 70. According to table 2, Fusarium sp. is sixth (2.4%). Check and change the sentences discussing Fusarium.
Response. The sentence has been corrected for a better understanding, and the discussion has been clarified.
Comment 71. Are you sure? Table 1. Different dissimilarity algorithms used in the study. Check!!
Response. The mistake was corrected.
Comment 72. saprophytic -> saprotrophic
Response. The mistake was corrected.
Comment 73. high-through put -> high-throughput
Response. The mistake was corrected.
Comment 74. fungi -> fungal
Response. The mistake was corrected.
Summary: As a result of the revision, the errors and suggestions marked were carried out. As well as this, the introduction was improved with new sentences and the material and methods section was clarified. Results and the figures related to were improved and the discussion and conclusions were revised, adding new sentences and references to improve the main text. Finally, an English revision was carried out in order to correct the grammar and the whole manuscript was revised to correct the use of the abbreviations sp. and spp. All the comments and suggestions received were highlighted and all the modifications built a new version of the manuscript. Any changes related to The English revision can be supervised and easily viewed by the editors and reviewers with “Track Changes” function”.

Round 2
Reviewer 2 Report
Dear authors and editors,
The manuscript is very much improved according to my suggestions and I find it suitable for publication in Forests journal now.
Best, Reviewer